# Acceptability and Tolerability of Extended Reality Relaxation Training with and without Wearable Neurofeedback in Pediatric Migraine

**DOI:** 10.3390/children10020329

**Published:** 2023-02-09

**Authors:** Mark Connelly, Madeline Boorigie, Klanci McCabe

**Affiliations:** 1Children’s Mercy Kansas City, Division of Developmental and Behavioral Health, University of Missouri-Kansas City School of Medicine, Kansas City, MO 64108, USA; 2Protara Therapeutics, New York, NY 10010, USA

**Keywords:** pediatric, migraine, pain, virtual reality, relaxation therapy, neurofeedback, wearable

## Abstract

*Objective.* To determine the acceptability of using extended reality (XR) relaxation training as a preventive treatment for pediatric migraine. *Methods.* Youths aged 10–17 years old with migraine were recruited from a specialty headache clinic and completed baseline measures evaluating their vestibular symptoms and attitudes about technology. The patients were then instructed in three XR-based relaxation training conditions (fully immersive virtual reality with and without neurofeedback, and augmented reality with neurofeedback), in counterbalanced order, and completed acceptability and side effect questionnaires after each. The patients also took XR equipment home for one week to use for relaxation practice and again completed the measures about their experience. The acceptability and side effect data were compared against predetermined acceptable thresholds and were evaluated for their association with the participant characteristics. *Results.* The aggregate acceptability questionnaire scores exceeded our minimum threshold of 3.5/5, with the two fully immersive virtual reality conditions preferred over augmented reality for relaxation training (*z* = −3.02, *p* = 0.003, and *z* = −2.31, *p* = 0.02). The endorsed side effects were rated by all but one participant as mild, with vertigo being the most common. The acceptability ratings were not reliably associated with age, sex, typical hours per day of technology use, or technology attitudes, but were inversely related to the side effect scores. *Conclusions*. The preliminary data on acceptability and tolerability of immersive XR technology for relaxation training among youths with migraine supports further intervention development work.

## 1. Introduction

Migraine is a common neurological disease, often beginning in childhood, that is one of the leading causes of disability worldwide [1,2]. The effective preventive treatment of migraine attacks in childhood is critical to prognosis, as each migraine increases the risk of migraine episodes becoming progressively more frequent and more disabling [3,4]. Migraine headaches are thought to be triggered by a period of dysfunctional sensory processing and the associated neurovascular changes that may be incited as part of the brain’s stress response [5,6]. Nonpharmacological treatments that promote the regulation of the stress response are therefore considered to have an important role in effective migraine headache prophylaxis. The importance of the preventive role of nonpharmacological treatments for migraine is augmented by evidence that pediatric migraine prevention medications are not reliably more beneficial than a placebo, but these medications are associated with higher rates of adverse events (e.g., fatigue, weight loss, paresthesia, mood changes) and can become costly if ineffective [7,8,9].

Relaxation training, with or without biofeedback, is a behavioral approach for supporting effective stress regulation that has been shown in some studies to improve the outcomes for patients with migraine [10,11,12]. However, accessing behavioral treatments for headache can be challenging when the treatment involves regular clinic visits with a healthcare provider [13]. Additionally, child and adolescent adherence for starting or regularly practicing relaxation treatment is variable and often suboptimal, thereby diminishing the potential efficacy of this intervention approach [14]. Matching the intervention format and design to patient interests can favorably modify attitudes about behavioral treatments and thus potentially improve the adherence with implementing treatment components [15].

Innovation, reduced costs, and the increased home use of technologies such as extended reality (XR) and wearable sensors create new possibilities to engage children and adolescent “digital natives [16]” in self-administered relaxation training and to potentially augment adherence and efficacy. In XR applications, users are presented with the ability to perceive and interact with entirely virtual environments (immersive virtual reality) or with virtual objects that are superimposed onto users’ view of their current physical environment (augmented and mixed reality). Data streams from a wearable biosensor can also be integrated in real-time to modify the objects seen and heard in the virtual environment once a target biosensor data threshold is approached. The integration with XR of data streams from wearable devices, such as brainwave data from a portable electroencephalogram headset, offers a novel approach to home- or clinic-based biofeedback/neurofeedback training. The augmentation of XR-based relaxation training with neurofeedback reinforcement may help children and adolescents prevent migraine episodes by shaping their skill at regulating the neural mechanisms thought to be involved in generating these episodes [17,18,19].

The most common clinical application of XR in pediatrics to date has been with medical procedures [20,21]. Virtual reality has analgesic and anxiolytic benefits in this context, likely through the mechanism of effectively engaging attention with pleasant sensory stimuli and modulating the autonomic, affective, and evaluative pain signaling pathways [22,23]. Studies have shown virtual reality to be perceived positively by youths when used for distraction during medical procedures [20,24]. However, the potential for the *prospective* use of XR-based relaxation interventions to manage recurring pain conditions such as migraine in children and adolescents is unknown. Some symptoms commonly associated with migraine, including vestibular symptoms (e.g., dizziness) and increased sensitivity to lights, sounds, and touch around the head, may also impact the viability of XR-based interventions in the migraine population. Thus, data on acceptability are first needed to gauge the potential of XR-based relaxation training as a preventive treatment option for youths with migraine.

To this end, the primary objective of the current study was to preliminarily determine the acceptability of XR relaxation training in youths with migraine. This work was completed as an initial step (Phase Ia [25]) toward informing the development of a XR-based preventive treatment protocol. We hypothesized that youths with migraine would appraise XR to be a satisfactory and acceptable means of training in relaxation skills with minimal adverse effects. We also sought to explore whether the patient acceptability ratings would vary as a function of whether neurofeedback with a wearable device is integrated, as a function of XR type (fully immersive virtual reality with a mounted headset versus augmented reality with a tablet), or by certain patient characteristics (age, sex, technology attitudes, and presence of vestibular symptoms).

## 2. Materials and Methods

### 2.1. Participants

Children and adolescents were recruited into the study from the specialty headache clinics associated with a dedicated pediatric hospital in Kansas City, MO. The screening and recruitment activities were completed between September 2021 and June 2022. Patients were considered eligible to participate if: they were between the ages of 10 and 18; had an established diagnosis of migraine, with or without aura, based on the criteria from the third edition of the International Classification of Headache Disorders [26]; were able to speak and read English; and had access to a mobile device at home (for pairing with the XR equipment during the home trial phase). Patients who had a documented cognitive or developmental issue that would preclude their understanding and participating in the study procedures were excluded. The minimum age of 10 years old was selected to ensure the adequate understanding of the survey questions, the fit of the devices, and the capacity for the autonomous use of the XR equipment. A sample size of 15 was targeted to include a range of participants’ perceptions while enabling an equal number of times for each of the three studied XR intervention conditions to be administered first, second, and third during in-clinic testing [27].

### 2.2. Procedures

The Institutional Review Board (IRB) of the Children’s Mercy Kansas City approved the study procedures (IRB #01225, initial approval date 27 March 2020). Patients scheduled for follow-up visits in a specialty headache clinic were prescreened by a research coordinator and then approached during a clinic visit to discuss the study. Assent to participate was obtained from patients ages 10–17 years old, and informed consent was obtained from their legally authorized representative.

Following the consent procedures, the participants completed the baseline questionnaires electronically, on a tablet via REDCap [28,29]. The participants were then trained by the research coordinator on the use of one of the three XR conditions: (1) immersive virtual reality without biofeedback (VR−bio); (2) immersive virtual reality with biofeedback (VR+bio); and (3) augmented reality with biofeedback (AR+bio). Table 1 provides a description of the relaxation content used for each of these options. The Pico G2 4K all-in-one counterweighted headset was used for the virtual reality hardware. This headset weighs 470 g and has an adjustable head strap suitable for child/adolescent head sizes. The headset has integrated speakers and an LCD display with a resolution of 3840 × 2160 and refresh rate of 75 Hz. For the conditions with neurofeedback (VR+bio, AR+bio), the BrainLink Lite wearable EEG headband from Macrotellect was used. This device is a wireless lightweight (40 g) band worn over the forehead and underneath the VR headset (when used together). The headband contains 3 dry forehead electrodes (frontal pole/FP1 EEG electrode, ground, and reference) to transmit the brainwave data via a Bluetooth connection for near real-time processing in the extended reality software application.

The participants remained seated in a chair away from walls in the clinic room during the XR procedures. For each of the three XR conditions, the participants were first shown a printed description of three content options and asked to choose one to try based on their preference (e.g., underwater scene, solar system scene). The content options had been curated from a commercially available app library (Healium) by the lead author (MC) for appropriateness for our study sample age range and for having minimal rapidly moving graphics. The content selected by the participant was then loaded into the custom XR software by the research coordinator. The participants were asked to relax and try to engage with the content until the scene ended (range of 4.5–7 min), but to let the coordinator know if they wanted to end early for any reason. Immediately upon completion, the patients were asked to complete a survey about their experience. Identical procedures were repeated two more times until the participant had tried all three of the XR conditions (VR−bio, VR+bio, AR+bio). The order in which the participants completed the three XR conditions was counterbalanced using replicated Latin Squares. There was a 5 min rest period between each condition.

The participants were then given the option to take home their choice of one of the three XR conditions for one week of home-based relaxation practice. The choice to do a home trial of XR for relaxation training was documented as one indicator of acceptability. The participants were encouraged to use the equipment to complete one of the relaxation training sessions at least a few times during the trial week. Device and software features beyond those used for the relaxation training content were locked during home use. After completion of the home trial, the participants were sent a link to complete an electronic follow-up survey about their experience. The participants also were provided with a prepaid shipping label and box to return the equipment at the end of the home trial period. The participants did not receive any payment for completing the study. All of the hardware were cleaned between participant uses by a portable plasmaclustering device (AirScout by TechUV, Lawrence, KS, USA) installed in a custom cabinet. 

### 2.3. Measures

#### 2.3.1. Baseline Measures 

Information on the participant demographics (age, biological sex, grade, in school, race, and ethnicity) and the elapsed time since the most recent headache (ranging between “currently have a headache” and “more than 30 days ago”) was obtained using a questionnaire created for this study. Headache diagnosis, as documented by the patient’s headache specialist according to the ICHD-3 criteria, was obtained from the electronic health record. Based on our expectation that vestibular symptoms may impact the tolerability of the XR conditions, the data on recent history of vestibular symptoms were obtained using the Pediatric Vestibular Symptom Questionnaire (PVSQ) [30]. The PVSQ asks the participants to report the frequency, over the past month, that they experienced vestibular symptoms (e.g., “A feeling that things are spinning or moving around”) using a 4-point scale (“never” to “most of the time”). The scores on the PVSQ were used to describe the range of vestibular symptoms in the sample and to explore the extent to which the occurrence of vestibular symptoms is associated with XR acceptability. Scores greater than 0.68 out of 3 on the PVSQ are interpreted as clinically elevated [30]. The internal consistency reliability (Cronbach alpha) on this scale for the current sample was α = 0.86.

Additionally, questions on participants’ perceptions about and use of technology were asked as part of the baseline questionnaire. The participants were asked about the types of devices they own and use regularly (e.g., smartphone, laptop, tablet, game console, wearable device), their prior experience with extended reality, the hours they spend using technology each day, and their general comfort with technology (0–100 visual analog scale ranging between “not at all comfortable” and “very comfortable”). The participants’ attitudes toward technology were also assessed using items from the attitude subscale of the Media and Technology Usage and Attitudes Scale [31]. For this scale, the participants were asked to indicate their agreement on a 5-point scale (“strongly disagree” to “strongly agree”) regarding evaluative statements about the value of technology (e.g., “Technology will provide solutions to many of our problems;” “New technology makes people waste too much time”). The internal consistency reliability of this scale for the current sample was α = 0.60. 

#### 2.3.2. In-Clinic Post Experience Surveys

To assess the perceived ease of use, utility, and satisfaction following the trial of each XR condition, a 12-item acceptability questionnaire was developed, comprised of items adapted from the mHealth App Usability Questionnaire and from the User Experience in Immersive Virtual Environments Questionnaire [32,33]. The sample items include, “This system would be useful for my health and well-being” and “I found that this virtual environment was lame (reverse-scored).” The participants were asked to respond to each item based on their experience using a 5-point agreement scale (“strongly disagree” to “strongly agree”). The internal consistency reliability on this scale for the current sample was α = 0.92. 

The Cybersickness Symptoms Questionnaire (CSQ) was used to assess the potential side effects (“virtual reality induced side effects”/VRISE) associated with each XR condition [34]. The participants were asked to rate, on a 4-point scale (“none”, “slight”, “moderate”, “severe”), if they experienced any of 8 symptoms, such as nausea, eyestrain, and dizziness, while trying the XR condition. The internal consistency reliability on this scale for the current sample was α = 0.86. The responses were dichotomized (“none” versus “slight/moderate/severe”) for some of the analyses.

The patients were additionally asked to provide responses to open-ended questions about what they perceived to be the most positive and negative parts of each XR condition, and if they had any suggestions for the improvement of each condition.

#### 2.3.3. Home Trial Post Experience Survey 

The adherence and barriers of home use of the XR equipment was assessed using a survey developed for this study. The participants were asked on how many days they used the XR for relaxation training and for how long they used it, on average, each time. If applicable, they were also asked to select the main reason for not using the device on days it was not used (e.g., equipment was not working, too busy with other things, forgot about it).

The same questionnaires used to assess the XR acceptability and side effects during the in-clinic procedures were repeated following the completion of the home trial. Additionally, the patients were again asked to provide responses to open-ended questions about what they perceived to be the most positive and negative parts of the XR condition they tried at home, and if they had any suggestions to improve the system they had tried.

### 2.4. Analyses

Descriptive statistics (frequency counts, estimates of central tendency and variability as applicable) were used to summarize the data on participant demographics, technology familiarity and attitudes, and scores on the PVSQ. For the in-clinic testing, descriptive statistics were used to summarize the data from the acceptability and side effects questionnaires for each of the three XR conditions. A planned non-parametric test for dependent samples (Wilcoxon Signed-Rank Test) was used to compare the acceptability and side effect scores between pairwise combinations of the three XR conditions. The effect size (*r*) for the Wilcoxon Signed-Rank Test was calculated as *r* = Z/(√N) [35]. We also evaluated the associations between the baseline variables (i.e., demographics, technology comfort, vestibular symptoms) and the acceptability questionnaire scores using correlation analyses (Pearson product-moment [r] or point-biserial [r_pb_], as applicable). 

For the home trial, the proportion of patients agreeing to try an XR condition at home was calculated as one indicator of acceptability. Descriptive statistics also were used to summarize the data from the acceptability and adverse effects questionnaires administered following the home trial, and for the home adherence and barriers items.

Our predetermined criteria for determining adequate acceptability and tolerability were as follows: (a) mean of agreement scores on the acceptability questionnaire >=3.5 (indicating at least moderately positive overall acceptability for the sample) for the in-clinic and home testing; (b) mean of scores on the adverse effects questionnaire <=1.5 (indicating modest to no experience of discomfort during use of the XR equipment); and (c) percentage of participants opting in and reporting repeated use of XR-based relaxation sessions during the home trial >=75%.

## 3. Results

### 3.1. Participant Characteristics

Of the 60 patients who were pre-screened for eligibility and approached by the study coordinator during a clinic visit, 24 patients (40%) were not interested in participating in a study. An additional 21 patients (35%) were interested in participating but did not have time to complete the consent and study procedures on the given day and would not be returning to the clinic until after our enrollment window. The remaining sample of 15 participants formed the study sample. The unenrolled sample did not systematically differ from the enrolled sample in known demographics (mean age, sex, race, and ethnicity). 

The demographics of the enrolled sample are presented in Table 2. The participants ranged in age between 10–17 years old (8 females, 7 males). The participants predominantly identified as White and non-Hispanic. Most of the patients (9/15, 60%) either had a current headache or had a headache in the past 24 h. One-third of the sample (5/15, 33.3%) reported prior experience with XR.

The participants generally reported high comfort with technology based on a 0–100 visual analog scale (M = 81.1, SD = 36.0, range 36–100). The scores on the technology attitudes scale clustered toward the middle and ranged between 1.6–3.8/5 (M = 2.96, SD = 0.56). The scores on the baseline PVSQ ranged between 0.33 to 2.67 (M = 1.16, SD = 0.71), with 9/15 (60%) patients having vestibular symptom scores in the elevated range [28].

### 3.2. In-Clinic Testing Results

Table 3 summarizes the responses to the in-clinic acceptability testing questionnaires, overall and as a function of the XR condition. The mean acceptability score, aggregated over all the XR conditions and participants, exceeded our defined minimum acceptable threshold of 3.5 (M = 3.82, SD = 0.47, range between 3.08–4.61). Both VR−bio (Mdn = 4.08) and VR+bio (Mdn = 3.75) were statistically favored, relative to AR+bio (Mdn = 3.58), *z* = −3.02, *p* = 0.003, *r* = 0.78, and *z* = −2.31, *p* = 0.02, *r* = 0.60, respectively. The VR−bio condition also generally had higher acceptability scores than VR+bio, *z* = −1.79, *p* = 0.07, *r* = 0.46. The percentages of patients with acceptability scores >=3.5 was 93.3% for VR+bio, 80% for VR−bio, and 60% for AR+bio. As shown in Table 2, the acceptability items with the strongest agreement overall included: “Overall, I am satisfied with the system”; “It was simple to use this system”; and “I enjoyed being in the virtual environment.”

Table 4 shows the correlations of the baseline variables with the total scores on the acceptability measure. The acceptability scores tended to be higher for younger participants, males, those with less vestibular symptoms, those reporting fewer hours of daily technology use, and those who rated higher comfort with technology. 

For the VR−bio condition, the scene most selected by the participants for viewing was the underwater cave (n = 7, 46.7%), followed by the beach scene (n = 5, 33.3%) and the forest and waterfall scene (n = 3, 20%). For VR+bio, the most selected scene was the snow and streaming water scene (n = 9, 60%), followed by the rainfall/storm scene (n = 4, 26.7%) and the waterfall scene (n = 2, 13.3%). Finally, for the AR+bio condition, the most selected scene was the solar system scene (n = 10, 66.7%), followed by the blooming flowers scene (n = 3, 20%) and the hatching butterflies scene (n = 2, 13.3%). 

The positive open-ended comments for each XR condition generally centered on the following themes: (a) the ability to be relaxed (e.g., “It was relaxing and I felt great after using it” [VR+bio]; “I like this because it helps with my breathing” [VR−bio]; “Feeling in control of my emotions” [AR+bio]); (b) enjoyable content (e.g., “That you were in a cool cave and that it felt like I was flying” [VR−bio]; “Interesting visuals, background music made it a little easier to get immersed” [VR+bio]; “I enjoyed how interactive it was and how the sun is controlled with brain waves” [AR+bio]); and (c) ease of use (e.g., “I think the most positive points were how cool and fun/easy to use” [VR−bio]). The negative comments and suggestions for improvement centered on: (a) some of the visual and audio elements (e.g., “Maybe don’t make the star thing flash ‘cause it made me feel dizzy” [VR−bio]; “The green line [that represented the target for relaxation during neurofeedback conditions] was a bit distracting” [VR+bio]; “Why is the sun in the room—why? I can’t focus because I’m trying not to laugh” [AR+bio]); (b) level of interactivity (e.g., “I feel like more interactivity is needed for it to be relaxing and for that feeling to last” [VR+bio]; “Maybe make it a little more interactive but it’s also great the way it is” [AR+bio]); and (c) the interface setup (e.g., “I didn’t like the feel of the headband and goggles” [VR+bio]; “Holding the large iPad up constantly to view the solar system got uncomfortable” [AR+bio]). 

### 3.3. Home Trial Results

When given the choice to take the XR equipment home to practice relaxation, all of the patients (15/15, 100%) expressed a desire to do so. Only one participant selected the AR+bio condition; the others selected a fully immersive VR condition (VR−bio, n = 12, or VR+bio, n = 2). The participants reported using the equipment for relaxation training on a median of 5/7 days during the home trial (range 1–7 days). The reported duration of use per time that the equipment was used for relaxation training was 6–10 min, which closely corresponds to the length of the relaxation training content options available on the XR application. Five participants (33%) reported a duration of use of greater than 10 min per use. The most common reason selected for not using the XR equipment on a given day was “getting too busy with other things” (selected by 4/15, 27%) or “forgetting about it” (selected by 3/15, 20%); one participant also reported challenges with getting the equipment to work. The number of days of reported use of the XR equipment during the home trial was moderately inversely related to participant age *(r* = −0.46, *p* = 0.08) and baseline PVSQ score (*r* = −0.44, *p* = 0.10). 

Table 3 also summarizes the responses to the acceptability testing questionnaire completed following the home trial. The mean acceptability score associated with the home trial was comparable to that during the clinic testing and exceeded our minimum threshold of 3.5 (M = 4.01, SD = 0.48, range between 3.1–4.8). The proportion of patients with acceptability scores ≥ 3.5 during the home trial was 93%. The overall acceptability ratings for the home trial were inversely associated with the baseline PVSQ scores (*r* = −0.53, *p* = 0.04). The positive open-ended comments from the home trial centered on the themes of effectiveness for achieving relaxation and ease of use (e.g., “You can do it at home without distractions. There was also a lot you can chose from”; “They were fun and made me calm down, and whenever I used it, I got a sense of well-being”; “It calmed me and made me feel like I was relaxing; I would definitely use it again”). Comments for improvement centered on some of the visual elements (e.g., “Umm, I would say the picture quality was a little distracting on some of the videos”) and the hardware (e.g., “The headband and headset were a little uncomfortable to wear at the same time”). 

### 3.4. Side Effect Results

#### 3.4.1. Side Effects during In-Clinic Testing

Figure 1 displays the side effect items that the participants endorsed as experiencing at least “slightly” while using an XR condition in clinic. The mean aggregate score for the side effects experienced was below our predefined acceptable maximum threshold of 1.5 (M = 1.34, SD = 0.43, range 1–2.38). The total side effect scores were comparable during VR−bio (M = 1.34, SD = 0.47), VR+bio (M = 1.37, SD = 0.45), and AR+bio (M = 1.29, SD = 0.45); there were no statistically significant differences in the side effect scores between VR+bio and VR-bio (z = −0.24, *p* = 0.81, r = 0.06), between VR+bio and AR+bio (z = −1.47, *p* = 0.14, r = 0.38), or between VR−bio and AR+bio (z = −0.54, *p* = 0.59, r = 0.14). The most frequently endorsed side effect across the conditions was vertigo, and the least was nausea. The side effect scores were positively associated with the presence of baseline vestibular symptoms, r = 0.69, *p* = 0.01. There was no statistically reliable (*p* < 0.05) difference in the side effect scores as a function of the choice of XR content. However, the scene with the highest average side effect score included relatively more moving objects (a hatching butterflies scene for AR+bio), and the scene with the lowest side effect score had relatively minimal movement (a beach scene for VR−bio). The mean acceptability ratings across XR conditions used in the clinic setting were inversely related to the side effect scores (r=−0.55, *p* = 0.03, 95% CI −0.83 to −0.05); those who reported experiencing more virtual reality-induced side effects reported lower acceptability ratings. 

#### 3.4.2. Side Effects during the Home Trial

Figure 1 also displays the side effect items that the participants endorsed as experiencing at least “slightly” while using an XR condition for relaxation training at home. The most frequently endorsed side effects during the home trial included difficulty focusing (endorsed by n = 5, 33%) and eyestrain (also endorsed by n = 5, 33%); the least commonly endorsed were nausea, blurred vision, and dizziness. The total scores for side effects with repeated use during the home trial remained below our acceptable threshold of 1.5 (M = 1.42, SD = 0.53, range 1–3.1) and, again, were highly associated with the presence of baseline vestibular symptoms, *r* = 0.85, *p* < 0.01. Most of the participants had total side effect scores in the range of “none” and “slight” side effects (14/15, 93%). One participant reported side effects of moderate intensity (dizziness and headache) and cited these side effects as a reason for only using the XR equipment once for home relaxation practice. The acceptability ratings completed for the home trial were strongly inversely correlated with the total side effect cores from the home trial, *r* = −0.78, *p* < 0.01.

## 4. Discussion

Behavioral interventions that teach ways to self-regulate the physiological systems implicated in migraine pathophysiology may be instrumental for migraine headache prevention [12,13,36], but youths’ adherence to home practice of these strategies is often suboptimal. The current study was completed to determine the extent to which learning relaxation skills through immersive or augmented virtual reality could be a viable behavioral intervention option for youths with migraine. We thought that XR would be viewed as an engaging and acceptable platform for shaping relaxation-based self-regulation skills for youths with migraine, thereby having potential value for being developed further into a personalized migraine prevention protocol. The results of the current preliminary study suggest that XR-based relaxation methods were viewed positively and tolerated by most youths with migraine who were willing to enroll in the study, regardless of age, sex, hours already spent per day with technology, and technology attitudes and comfort. All of the participants desired to try XR-based relaxation training prospectively at home, and most reported repeated practice over the home trial. The acceptability ratings were found to be somewhat lower for patients with frequent vestibular symptoms at baseline or who experienced some side effects during the use of the XR equipment. Fully immersive virtual reality was also found to be preferred as a format for relaxation training over augmented reality. 

The finding from our sample that XR was generally viewed positively and was thought to help with relaxation and well-being is consistent with other pediatric use cases of virtual reality. Research on the pain applications of XR, to date, has mostly been limited to use of immersive VR for children and adolescents undergoing medical procedures [20,21,24,27]. These studies have indicated that youths across a wide age range find VR to be an enjoyable experience that successfully distracts them from a medical procedure, although success with distraction has not always translated into reductions of the targeted outcome of pain ratings or distress in these studies [24,37,38,39]. 

We did not know whether youths with migraine would view XR interventions similarly as other pediatric samples given the enhanced sensitivity to motion and sensory inputs (touch, vision, sound) that are common in migraine. Indeed, the need to capture side effects during studies of XR has been emphasized in some reviews due to the potential for visually-induced motion sickness that may be more pronounced in some pediatric populations or with some types of VR content [21]. The experience of XR side effects is largely unknown in pediatric samples due to this information rarely being systematically tracked and reported. We found that although the reported side effects were mild for most of the participants during relatively brief “doses” of the studied XR conditions, the experience of ostensibly XR-induced side effects did result in lower intervention acceptability ratings. Moreover, some of the symptoms endorsed as occurring during XR use (e.g., headache, nausea) are what the intervention approach would be intended to reduce rather than incite. However, the participants were asked to indicate if they experienced any of the symptoms on a list during the period of XR exposure, rather than being asked if the symptoms started or worsened during the period of XR exposure. As such, it may be that the participants endorsed a “side effect” because it was already being experienced at baseline. Indeed, the side effects most endorsed in this study are symptoms that are common with migraine, independent of any treatment exposure [40].

We also found that the baseline vestibular symptoms were highly correlated with the intensity of the side effects. Youths with a vestibular component to their migraine phenotype may have a greater sensitivity to moving elements in XR and, thereby, be more likely to experience dizziness, vertigo, or headache. As such, having intense or frequent vestibular symptoms is likely a contraindication for an XR-based relaxation intervention. Interestingly, however, the participants who endorsed baseline vestibular symptoms still engaged with and viewed the XR formats positively, even with the experience of side effects. Thus, although not evaluated here, XR might have potential for use in the controlled gradual exposure and desensitization to vestibular symptoms [41,42]. 

Research is comparatively scarce on the acceptability and utility of the ongoing use of XR applications to support the management of chronic health conditions. Through repeatedly transforming how patients perceive their bodies and sensory environment, the prospective use of XR in chronic health conditions may have a unique capacity to modify the cortical and other physiological patterns relevant to the symptoms experienced [21]. For example, some studies have evaluated the prospective use, in a clinic setting, of VR mirror therapy (using virtual representation of arms or legs) to shape the increased comfort with limb movement for certain pain conditions (e.g., Complex Regional Pain Syndrome). The limited available data on the prospective use of XR in children has indicated good patient acceptability in a clinic setting [43], although the impact on objective health outcomes has not been established. For the current study, we found that most of the children in our sample enjoyed the ongoing use of XR for relaxation practice in their home setting, typically doing a relaxation session most days over a one week trial. The experience of side effects with repeated use generally did not lead to discontinuation but was associated with lower overall acceptability scores. Difficulty focusing and eyestrain during the XR exposure at home were commonly endorsed as mild side effects, perhaps due in part to more distractions when trying to use the equipment at home and greater frequency or duration of use, respectively. However, blurred vision, dizziness, headache, and nausea were more infrequently endorsed relative to the in-clinic testing. Future research is needed to determine the frequency and duration of XR-based relaxation training, if any, that achieves a clinically important difference in youth migraine occurrence, while minimizing these side effects.

Adding biofeedback to virtual reality to shape and reinforce the acquisition of a target range on physiological variables has begun to be explored in some studies, primarily with adult participants [44]. Biofeedback (including neurofeedback) has a well-established evidence base for improving headache outcomes, but young patients may struggle to remain sufficiently motivated and engaged with traditional biofeedback training to achieve success [12,45,46]. Integrating biofeedback with XR may promote child and adolescent engagement in biofeedback-assisted relaxation training through attractive elements, such as novel multisensory representations of physiological signals, interactivity, and gamification [44,46]. Matching the physiological signal to be modified during XR-based relaxation training to the known pathophysiology of a condition, such as cortical excitability in migraine [17,47], also may provide a more robust physiological and therapeutic response relative to traditional biofeedback or other approaches to relaxation training [48,49]. For two of the XR conditions in the current study, we used the data on the level of relaxation from a consumer wearable EEG headband to induce real-time changes in the content participants were viewing and hearing. The participants generally viewed the integration of neurofeedback with immersive virtual reality positively, but the acceptability ratings were lowest when neurofeedback was paired with augmented reality viewed through a tablet device. This suggests that even with the relative awkwardness noted by a couple of the participants of wearing two devices on the head (a VR headset and EEG headband), the experience of fully immersive visual content was preferred to non-immersive content by the youths with migraine. This may be related to immersive VR allowing youths with frequent head pain to be more fully transported from their current setting to a calming and comfortable sensory environment for practicing relaxation. However, further research is needed to establish the actual incremental value of immersive VR with neurofeedback relative to other interventions for treating pediatric migraine.

Our study has several limitations to consider when interpreting the results. As it was a preliminary pre-efficacy study, completed as part of early phase intervention development work, the sample size was small and homogeneous; as such, the findings may not be generalizable beyond the current sample. Similarly, the acceptability findings are based on the perceptions of youths with migraine who were sufficiently interested in XR technology to agree to participate in the study. As such, the findings on the acceptability may be upwardly biased and not fully representative of the perspectives of youths with migraine. Additionally, our evaluation of the prospective use of XR for relaxation training in the home setting was limited to a short time frame, and the usage information during that time was based on self-report. Further research is needed that objectively monitors adherence over longer time periods and identifies the dose-dependency of any observed benefit in preventing migraine episodes. Additionally, although we aimed to systematically quantify the side effects as an indicator of the XR tolerability, our approach to measuring the side effects did not enable us to clearly differentiate between symptoms already experienced by the participant versus symptoms induced or worsened by XR exposure.

The extent to which the study findings are specific to the XR content and hardware used for the current study also is unknown. This limitation was partly mitigated by evaluating a variety of content and enabling the participants to have a choice of content, based on evidence that the personalization of content positively contributes to relaxation and engagement [50]. However, the generalizability of the findings across hardware options cannot yet be determined but may have implications for cost. For example, VR headsets that use an inserted smartphone for displaying content can cost less than 50 USD but may sacrifice features that help minimize visual discomfort and promote immersion for youths with migraine. Similarly, we do not know if using hardware that is increasingly already owned by youths (e.g., activity tracker or smartwatch) for biofeedback integration would be comparable in its acceptability or benefits to the wearable EEG headband used in the current study. The cost-benefit of (and adherence with) any XR approach to pediatric migraine prevention still needs to be established relative to other preventive treatments to justify clinical use.

Overall, the results of the current study preliminarily indicate that the use of XR technology, particularly immersive virtual reality with or without neurofeedback, does seem to have some promise for being an acceptable and engaging approach to relaxation training for a subset of youths with migraine. However, further work is needed on the customizing of intervention features to address the preferences and sensitivities of youths with migraine. Further, the comparative benefit/risk of a VR-based relaxation intervention for preventing migraine episodes is yet unknown and requires formal evaluation.

## Figures and Tables

**Figure 1 children-10-00329-f001:**
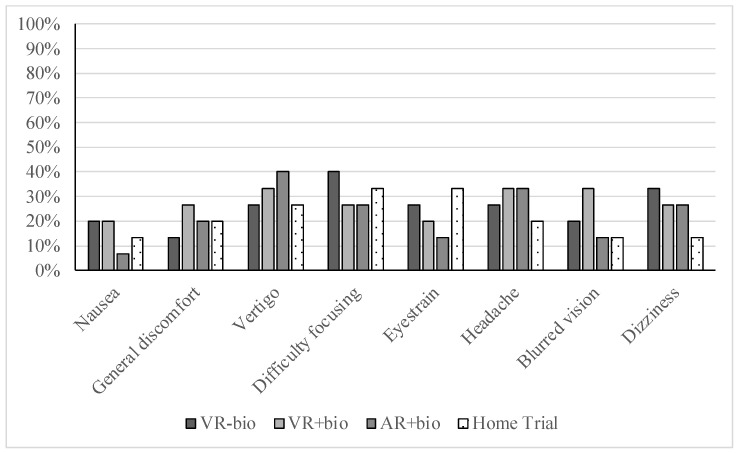
Percentages of participants reporting side effects during experience with XR. Shown in the figure are the percentages of the sample who endorsed experiencing the given side effect item in at least “slightly” during their experience with XR. Percentages shown for VR−bio (immersive virtual reality without integration of biofeedback), VR+bio (immersive virtual reality with integration of biofeedback from a wearable electroencephalogram headset), and AR+bio (augmented virtual reality with integration of biofeedback from a wearable electroencephalogram headset) are based on ratings during the in-clinic experience with XR-based relaxation training. Percentages shown for “home trial” are based on ratings from prospectively trying one of the XR conditions for relaxation training at home.

**Table 1 children-10-00329-t001:** Description of XR conditions used in the study.

Condition	Description	Hardware Used
(1) VR-bio: Immersive virtual reality relaxation training without use of a paired wearable sensor for biofeedback	Participants were instructed to relax and look all around (360 degrees) as desired while wearing a virtual reality headset. Virtual reality content included choice of: (a) an underwater scene; (b) a forest and waterfall scene; or (c) a beach scene. Audio of calming music and/or nature sounds, along with instruction in relaxed breathing, autogenic relaxation, or mindfulness, were simultaneously played through the headset.	(1) Pico G2 4K all-in-one virtual reality headset
(2) VR+bio: Immersive virtual reality with use of a paired wearable sensor for biofeedback	Participants were instructed to relax while wearing a virtual reality headset combined with a consumer electroencephalography (EEG) sensor headband. Virtual reality content included choice of: (a) a snow and streaming water scene; (b) a waterfall scene; or (c) a rainfall scene. Overlaid on the virtual reality imagery is a thin horizontal line representing the patient’s baseline EEG activity, along with a glowing firefly that moves up or down off the horizontal line depending on the user’s calmness level. Calmness level is calculated by an algorithm applied to near-real time brainwave frequency band data (delta, theta, low-alpha, high-alpha, low-beta, high-beta, low-gamma, and mid-gamma) from the single-channel EEG headband. Additionally, the virtual environment changes based on the brainwave data (e.g., rain stops, clouds clear, and a rainbow appears when brainwave data are interpreted to indicate success in sustaining reduced stress/calmness based on reduction in high beta/low gamma EEG activity). Audio instruction in relaxation strategies (relaxed breathing, autogenic relaxation, or mindfulness), along with calming music, is simultaneously played through the virtual reality headset.	(1) Pico G2 4K all-in-one virtual reality headset (2) Macrotellect Brainlink Lite EEG headband
(3) AR+bio: Augmented reality with use of a paired wearable sensor for biofeedback	Participants were instructed to wear the EEG headband while engaging with audio-visual content via an app on a tablet (iPad). Virtual content comprised the choice of (a) a butterfly scene; (b) a solar system scene; or (c) a flowers scene. This visual content viewed by users in the app is overlaid on top of live video input from the tablet’s camera. The virtual content changes depending on level of relaxation, based on processed calmness data from the EEG headband (e.g., flowers grow, the sun grows bigger and brighter, or butterflies start to hatch when brainwave data are interpreted to indicate success in sustaining reduced stress/calmness based on reduction in high beta/low gamma EEG activity). Audio instruction in relaxed breathing, autogenic relaxation, or mindfulness, along with calming music, is played as part of the content.	(1) Apple iPad tablet (2) Macrotellect Brainlink Lite EEG headband

**Table 2 children-10-00329-t002:** Baseline characteristics of the enrolled sample.

Variable	Sample Value
Sex, n (%)	
Male	7 (46.7%)
Female	8 (53.3%)
Race, n (%)	
White	14 (93.3%)
Black/African-American	1 (6.7%)
Ethnicity, n (%)	
Not Hispanic or Latino	14 (93.3%)
Hispanic or Latino	1 (6.7%)
Age, M ± SD (min - max)	12.9 ± 2.0 (10–17)
Grade in school, M ± SD (min – max)	7.6 ± 2.1 (4–11)
Diagnoses, n (%)	
Chronic migraine without aura	7 (46.7%)
Migraine without aura	4 (26.7%)
Migraine with aura	3 (20.0%)
Chronic migraine with aura	1 (6.7%)
Technology ownership, n (%)	
Smartphone	13 (86.7%)
Desktop computer	8 (53.3%)
Tablet	4 (26.7%)
Laptop	4 (26.7%)
Game console	2 (13.3%)
Wearable device	2 (13.3%)
Prior experience with XR, n (%)	
No	10 (66.7%)
Yes	5 (33.3%)
Hours per day of technology use, M ± SD (min to max)	6.87 ± 4.81 (0–21)

Notes: PVSQ = Pediatric Vestibular Symptoms Questionnaire; XR = Extended reality; M = Mean; SD = Standard deviation; min = minimum reported value on this variable, and max = maximum reported value on this variable.

**Table 3 children-10-00329-t003:** Summary of responses to acceptability items during the in-clinic and home experience with XR.

Item	VR-bio (M ± SD)	VR+bio (M ± SD)	AR+bio (M ± SD)	Overall (M ± SD)	Home Trial (M ± SD)
It was simple to use this system.	4.33 ± 0.49	4.13 ± 0.35	4.00 ± 0.38	4.16 ± 0.42	4.09 ± 0.54
The display quality was distracting. *	2.13 ± 0.83	2.20 ± 0.94	2.60 ± 1.06	2.31 ± 0.95	2.36 ± 0.81
I felt I controlled the situation.	3.73 ± 0.80	3.40 ± 1.18	3.33 ± 0.82	3.49 ± 0.94	4.00 ± 0.63
The experience gave me a sense of well-being.	4.07 ± 0.59	3.73 ± 0.80	3.53 ± 0.74	3.78 ± 0.74	4.18 ± 0.74
I enjoyed being in the virtual environment.	4.33 ± 0.62	4.07 ± 0.80	3.73 ± 0.80	4.04 ± 0.77	4.55 ± 0.52
I got tense in the virtual environment. *	2.07 ± 0.80	2.07 ± 0.80	2.60 ± 0.91	2.24 ± 0.86	2.18 ± 0.98
I feel confident I could learn skills using this type of program.	3.80 ± 0.94	3.33 ± 0.82	3.13 ± 0.83	3.42 ± 0.89	3.64 ± 1.03
I found this virtual environment was lame. *	1.60 ± 0.83	2.07 ± 0.88	2.53 ± 1.19	2.07 ± 1.03	2.09 ± 0.70
I liked the interface of this system.	4.07 ± 0.88	3.80 ± 0.78	3.60 ± 0.83	3.82 ± 0.83	3.82 ± 0.40
I would use the system again.	4.33 ± 0.62	4.13 ± 0.74	3.40 ± 0.74	3.96 ± 0.80	4.27 ± 0.65
The system would be useful for my health and well-being.	4.00 ± 0.66	3.67 ± 0.90	3.53 ± 0.74	3.73 ± 0.78	3.91 ± 0.70
Overall, I am satisfied with the system.	4.47 ± 0.67	4.20 ± 0.56	3.60 ± 0.83	4.09 ± 0.76	4.27 ± 0.47
Total acceptability score (mean of all items)	4.11 ± 0.49	3.84 ± 0.57	3.51 ± 0.62	3.82 ± 0.60	3.94 ± 0.44

Notes: Item responses are on a 5-point scale ranging between 1 (“Strongly Disagree”) and 5 (“Strongly Agree”). Data presented in this table are mean (M) ± standard deviation (SD). “Overall” refers to grand mean for the item across the three XR conditions tried in clinic. * These items were reverse-scored in calculating the total acceptability score.

**Table 4 children-10-00329-t004:** Correlation of variables with total scores on the acceptability questionnaire.

Variable	*r* (95% CI)	*p*
Age	−0.17 (−0.63 to 0.37)	0.54
Sex	−0.43 (−0.76 to 0.14)	0.14
PVSQ Total Score	−0.43 (−0.77 to 0.11)	0.11
Hours per day of technology use	−0.17 (−0.63 to 0.38)	0.55
Technology Attitudes Scale Score	−0.44 (−0.09 to 0.78)	0.10
Technology Comfort Score	0.22 (−0.33 to 0.66)	0.43

Notes: Point-biserial correlation (*r*_pb_) is shown for the association between sex and total acceptability score; all other *r* values are Pearson product-moment correlations. PVSQ = Pediatric Vestibular Symptoms Questionnaire.

## Data Availability

A deidentified study dataset can be obtained upon request by contacting the corresponding author.

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
