# Peer review of "Acceptability and Tolerability of Extended Reality Relaxation Training with and without Wearable Neurofeedback in Pediatric Migraine"

_children, 2023, doi:10.3390/children10020329_

Round 1

Reviewer 1 Report

Thank you for the opportunity to review this well-written pre-efficacy study to explore the acceptability of using extended reality relaxation training as a preventive treatment approach for pediatric migraine  While the current study definitely has its strengths and is in my opinion very well-written and of good quality, I believe there are some small issues that could be addressed to further improve the current paper.

Introduction

1. Line 35-39 Can the authors briefly state the costs and significant side-effects that are associated with the approved pediatric migraine prevention medications? That would strengthen justification for why we should engage in non-pharmacological preventive methods.

Methods

2. Line 92: Can the authors add justification for the chosen age range?

3. Line 103: “The Institutional Review Board (IRB) of Children’s Mercy Kansas City approved the 103 study procedures (IRB #01225).” Add date of approval.

4. Line 110: Participants were then trained by the research coordinator on use THE of one of the three XR conditions

5. Was there a break between switching from one to the other condition?

6. In all three groups, relaxation strategies were instructed, but were the same relaxation strategies instructed or did they differ between the groups?

7. Baseline measures: was any information collected on migraine aura, triggers, and symptoms (besides headache)?

Results

8. Line 237: The authors state that 21 patients (35%) were interested to participate but did not have time to complete consent and in-clinic study procedures the day they were approached. I was wondering why the research team choose to only include patients that wanted to participate on the same day as informed consent. Why did the team not give participants the time tot hink about whether they wanted to participate or not, and plan study participation during their next appointment.

Discussion

9. Can the authors elaborate on possible reasons for differences in experienced side effects between the in-clinic testing and home trial

10. Can the authors elaborate shortly on their thoughts concerning cost-benefit of preventively offering XR to migraine patients? I think it is an innovative approach and many patients will like this, but is this feasible for hospitals/medical settings to offer XR materials to all of their migraine patients?

Reviewer 2 Report

This manuscript described the acceptability and tolerability of extended reality relaxation (virtual reality and augmented reality) with and without wearable neurofeedback in pediatric migraine. This is a very small study, only 15 subjects of 60 prescreened subjects participated in the study. The study period is also very short, one-time in-clinical testing, and one-week home-setting testing. The home setting portion is not well controlled (for example, the XR methods can not be grouped into VR-bio, VR+bio, or AR+bio; the relaxation time is not controlled….). While authors recorded the side effects using XR, they did not discuss the implications of side effects on migraine. Many of the side effects observed in the study may trigger migraines, for example, vertigo (the most common side effect observed in this study). While authors did not record the efficacy of using XR  on managing migraines (not the aim of this study), they did observe as high as over 30% having the headache as a side effect, which is the opposite of the purpose of using XR therapy. The contents of XR used are also important, and authors did not discuss the effects of contents on the side effects of XR (or potential on managing migraine). Overall, authors should do a more thorough analysis of their data and critically discuss those data. It appears that this is just a pilot study, but the poor design of the study may not offer enough data to conclude the tolerability. I would recommend authors thoroughly revise the manuscript.

Round 2

Reviewer 2 Report

The revision is fine. While authors mentioned they did not differentiate the headache as a side effect induced by XR or the baseline experience in the discussion part, this is apparently the design limitation, and they may mention it again in the study limitations part. Other than that, I think it can be accepted for publication. 
